

# Creation of mutants by using centrality criteria in social network analysis

Savaş Takan

Department of Computer Engineering, Faculty of Engineering, Izmir Institute of Technology, Izmir, Turkey

## ABSTRACT

Mutation testing is a method widely used to evaluate the effectiveness of the test suite in hardware and software tests or to design new software tests. In mutation testing, the original model is systematically mutated using certain error assumptions. Mutation testing is based on well-defined mutation operators that imitate typical programming errors or which form highly successful test suites. The success of test suites is determined by the rate of killing mutants created through mutation operators. Because of the high number of mutants in mutation testing, the calculation cost increases in the testing of finite state machines (FSM). Under the assumption that each mutant is of equal value, random selection can be a practical method of mutant reduction. However, in this study, it was assumed that each mutant did not have an equal value. Starting from this point of view, a new mutant reduction method was proposed by using the centrality criteria in social network analysis. It was assumed that the central regions selected within this frame were the regions from where test cases pass the most. To evaluate the proposed method, besides the feature of detecting all failures related to the model, the widely-used W method was chosen. Random and proposed mutant reduction methods were compared with respect to their success by using test suites. As a result of the evaluations, it was discovered that mutants selected via the proposed reduction technique revealed a higher performance. Furthermore, it was observed that the proposed method reduced the cost of mutation testing.

## INTRODUCTION

Testing provides the means to check whether the pre-specified requirements have been met, and the correct outputs have been produced. We benefit from software testing to eliminate potential faults and to ensure error-free operation of the software (*Mathur, 2013*). To this end, because there are a number of text strategies, a large number of test cases need to be generated.

In computer science and particularly in software and hardware engineering, in order to develop and verify the software and hardware systems, formal methods are used to convert these complex systems into mathematical models. With the use of formal methods, it is possible to verify the reliability of a mathematically modelable design with empirical tests in a more comprehensive way (*Mathur, 2013*).

Corresponding author
Savaş Takan, savastakan@iyte.edu.tr

In software testing, formal method-based testing constitutes a popular research topic (*Lee & Yannakakis, 1996*; *Endo & Simao, 2013*). Because formal methods provide the opportunity to detect whether the model meets the requirements or not. Finite State Machines (FSM), Petri Nets, and UML are some of the formal methods. FSM, which is mainly used to model hardware and software systems, is a mathematical model which has discrete inputs and outputs. Various methods are proposed in the literature to form test suites in FSMs, the most widely used of which are Transition Tour (*Naito, 1981*), W (*Chow, 1978*), Wp (*Fujiwara et al., 1991*), UIO (*Sabnani & Dahbura, 1988*), UIOv (*Vuong, 1989*), DS (*Gonenc, 1970*), HSI (*Petrenko & Yevtushenko, 2005*; *Petrenko et al., 1993*; *Yevtushenko & Petrenko, 1990*) and H (*Dorofeeva & Koufareva, 2002*). In this study, to evaluate the proposed method, in addition to the feature of detecting all the errors in the model, the W method, was chosen because it is widely used.

Mutation testing is generally used to measure the effectiveness of tests or to design new software tests in the formal methods (FSM, Petri Net, and UML). In the mutation test, the original model is systematically mutated using certain error assumptions. The purpose here is to represent a potential programmer error. In this line, a small change (mutation) is made in the source code. These changes are based on well-defined mutation operators which imitate typical programming failures or which enable for highly successful tests to be established.

An example mutation process is given below:

$$if\,(a < b)\{\ldots\} \quad \rightsquigarrow \quad if\,(a > b)\{\ldots\}$$

Each modified version of a program, as in the equation above, is called a mutant. If a test case can detect a certain mutant, it is accepted that the mutant is killed by this test case. The Performance of a test suite is determined as per the killing rate of mutants by means of the mutation operators. Mutation analysis generally has many advantages; however, doing the analysis for huge models require too much time and space because of the high number of mutants. To avoid this situation, mutant reduction methods are used. Mutant reduction enables the analysis to be completed in a reasonable time and space by limiting the number of mutants.

Among the various mutant reduction methods, the most widely used one is random selection method which is based on a random sub-set selection of mutants. The random selection method is based on the assumption that mutants are identical in terms of their probability of occurrence. Within the frame of this assumption, the random selection method is a very practical mutant reduction method. However, this assumption is inaccurate since the situations where mutants are killed by the test suite is often not identical.

Although there are numerous studies in the literature related to mutant reduction, there are few related to FSM-based mutant reduction (*Fabbri, Maldonado & Delamaro, 1999*; *Maldonado et al., 2001*; *Fabbri et al., 1994*; *da Silva Simao et al., 2008*; *Li, Dai & Li, 2009*; *Petrenko, Timo & Ramesh, 2016*; *Timo, Petrenko & Ramesh, 2017*). Existing studies are generally focused on the mutation operators being required for mutant generation (*Pizzoleto et al., 2019*). In this study a different situation with a new FSM-based mutant reduction method was proposed.

In the study, it was first assumed that each mutant does not have equal value. Starting from this point of view, a new mutant reduction method was proposed by using the centrality criteria in social network analysis. It was assumed that the central regions selected in this frame are the regions from where the test suite passed the most. To evaluate the proposed method, in addition to the feature of detecting all errors relating with the model, W method was chosen because it is widely used. Random and proposed mutant reduction methods were compared with respect to their performances by using the test suite.

Social network analysis defines analysis of all types of structures as being correlated with one another. Social network analysis examines the social structure as a network of actors (nodes) and sets of relationships that connect actor pairs and examine the social structure and its effects. In short, social network analysis is a set of methods developed to analyze social networks. Social network analysis, originating from sociology and mathematics, is a method used by various disciplines nowadays. Beginning with the wide dissemination of computer usage by the end of the 1980s, social network analysis has become widespread for various reasons such as having easier access to big data sets, managing big data sets easier, and visualizing data related to social networks in different forms.

In social network analysis, to determine the most important actors and groups, some centrality criteria are used, the most common of which are degree centrality, eigenvector centrality, closeness centrality, and betweenness centrality. In addition to these, it is also possible to obtain information about centrality by using a clustering coefficient.

In the study, the regions from where the test cases pass the most were determined by using centrality criteria, a social network analysis method. The mutants were then generated using mutation operators including missing of transition (MOT), change of input (COI), change of output (COO), and change of next state (CONS). Afterwards, mutants created as per centrality criteria and randomly reduced mutants were evaluated by using test suite that had been created with the W method. The results were presented with graphical comparisons and with the ANOVA test. It was shown that the proposed mutant reduction methods could enable an important increase in performance.

Chapter 2 contains the literature on the subject. In the following section, Finite State Machine Testing. Then the W Method is examined. In the 'Mutation Texting' section, Mutation Analysis is mentioned in the study. Subsequently the main subject of the article is the creation of mutants using centrality criteria in social network analysis. Finally, the obtained data are tested and evaluated.

## LITERATURE REVIEW

In the last twenty years, there has significant increase in the number of studies related to mutant reduction (*Jia & Harman, 2010*; *Silva, de Souza & de Souza, 2017*; *Ferrari, Pizzoleto & Offutt, 2018*; *Papadakis et al., 2019*; *Pizzoleto et al., 2019*). In this section, pioneering works in the literature related to this subject are briefly explained.

*Acree et al. (1979)* proposed a random mutant reduction method based on formation of mutants as per a probability distribution which had been previously determined. Another

method proposed by *Acree et al. (1979)* was a higher order mutant reduction method a technique combining two or more number of simple mutants in order to create a single complex mutant.

Later, *Howden (1982)* worked on the weak mutation technique which controls whether a mutant is infected or not. If the mutant is infected, the mutant is promptly killed. On the other hand, *Woodward & Halewood (1988)* proposed a firm mutation technique, a form of the weak mutation technique.

*Krauser, Mathur & Rego (1991)* proposed the parallel execution technique which executes mutants with parallel processors and which reduces total period required to make mutation testing. *Marshall et al. (1990)* proposed data flow analysis as a mutant reduction method. In data-flow analysis, a program uses information related to data flow to decide which mutants will be produced.

*Sahinoglu & Spafford (1990)* proposed a technique that analyzes a test suite in order to score test cases as per their effectiveness in killing mutants and that eliminated the ineffective test cases. *DeMillo, Krauser & Mathur (1991)* used techniques related to a compiler in order to reduce mutants. *Mathur (1991)*, on the other hand, proposed the constrained mutation technique which selected a sub-set of mutation operators to be used.

*Untch (1992)* proposed a method that produces and executes mutants by embedding all mutants in a parameter program called meta-mutant. Accordingly, the meta-mutant is then compiled for quick execution. When it is operated, the meta-mutant takes a parameter that informs which mutant will be operated.

*DeMillo & Offutt (1993)* proposed a technique using information related to the program control flow focusing on executive features to define the branches and commands helping to determine which structures are related to the production and execution of mutants. *Offutt, Rothermel & Zapf (1993)* also conducted studies related to the selective mutation technique.

*Weiss & Fleyshgakker (1993)* reduced the number of mutants to be executed by determining the mutant classes that act in a similar way. *Barbosa, Maldonado & Vincenzi (2001)* tried to determine a fundamental set of mutation operators by applying privatized procedures.

*Adamopoulos, Harman & Hierons (2004)* used evolutionary algorithms to reduce the number of mutants, to reduce the number of test cases, or to define equivalent mutants. *Untch (2009)* proposed a single mutation operator which covered the program the most and which produced the least number of mutants. On the other hand, *Fraser & Zeller (2012)* used the execution traces of the original program and some mutants to decide which of the remaining mutants should be executed.

*Aichernig, Jöbstl & Kegele (2013)* recommended the model-based mutant reduction method. This technique changes models of the program. Afterwards, test suites are created from these mutants, and these test suites are used to kill the mutants.

*Gligoric, Jagannath & Marinov (2010)* proposed a state-based analysis technique. This technique compares different mutants, and when two mutants cause the same mutation, meaning that when same execution path is observed, only one of them needs to be executed and the other one can be eliminated.

*Ammann, Delamaro & Offutt (2014)* proposed a technique eliminating unnecessary mutants by applying mutant subsumption and dominator mutant concepts.

Mutation analysis studies related to FSM are limited, and those that exist mainly examine the mutation operators that are used to produce mutants (*Fabbri et al., 1994*; *da Silva Simao et al., 2008*; *Li, Dai & Li, 2009*; *Petrenko, Timo & Ramesh, 2016*; *Timo, Petrenko & Ramesh, 2017*). Mutant reduction has been evaluated in many studies (*Jia & Harman, 2010*; *Silva, de Souza & de Souza, 2017*; *Ferrari, Pizzoleto & Offutt, 2018*; *Papadakis et al., 2019*; *Pizzoleto et al., 2019*), but, to our knowledge, there are no studies regarding mutant reduction in relation to FSMs. The purpose of this study is to reduce mutants and to make mutation analysis more practical and effective for FSMs. The method we propose uses social network analysis centrality criteria. This study is based on the assumption that the states and transitions detected with the help of centrality criteria are the parts that are most likely to cause errors. When a mutant is created with the states and transitions being selected with social network analysis centrality criteria, while the number of mutants decrease, the failure detection rate per mutant increases. The proposed mutant reduction methods revealed important performance increases in comparison to a random mutant reduction method.

## FINITE STATE MACHINE TEST

Finite State Machine (FSM) is a mathematical model with discrete inputs and outputs used for the modeling of software and hardware systems. It is possible to consider FSM an abstract machine that can be present in one of a finite number of states. Many structures such as text editors, compilers, and synchronous sequential circuits can be modeled by FSMs. Digital computers can also be considered as systems that comply with this model. For this reason, FSM is a very widely used model in computer science and engineering (*Belli et al., 2015*; *Fragal et al., 2019*; *Damasceno, 2016*).

A finite state machine is usually expressed with the tuple $M = \langle I, O, S, f, g, s_0 \rangle$, where $I$ is a finite set of input symbols, $O$ is a finite set of output symbols, $S$ is a set of finite states, $f : S \times I \rightarrow S$ is the transition function which determines the next state, $g : S \times I \rightarrow O$ is the output function and $s_0$ is the initial state of the machine.

Figure 1 shows a two-input and a one-output FSM with four states and seven transitions. We will use it as the running example in the rest of the article.

FSMs are widely used in software testing, and one of the most important issues in software testing is related to the formation of test suites. In the literature, there are many test suite generation methods such as W, Wp, UIO, UIOv, DS, HSI, and H. Before introducing techniques for FSM-based testing, two definitions should be differentiated:

**Test Cases:** A set of input values developed for a specific purpose or test conditions such as running a specific program path or verifying compatibility with a requirement, with a set of prerequisites to be executed before the test, or with the expected results and conditions as a whole.

**Test Suite:** A set of test cases being created to test a system or a component.

Another important point is related to the increase in the varieties and dimensions of software which necessitates better testing tools within the framework of various success

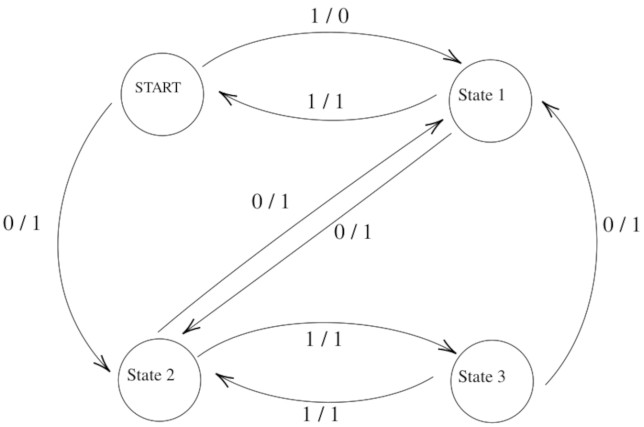

**Figure 1 Example Finite State Machine (FSM).**

criteria such as failure detection rate, failure detection capability, test suite size, test case lengths, and time spent in testing, with regards to the tests.

As mentioned above, in the study regarding the testing of finite state machines, W method is preferred because the W method can produce a test set that can detect all potential failures, meaning that it can detection in order to kill all potential mutants. Thus, the W method has a structure that can catch all mutants which can be formed. In the study, mutants randomly created and mutants reduced by using centrality criteria in social network analysis were compared with respect to the context of the test set being formed by the W method.

## W METHOD

W is one of the most common methods for generating test cases for FSMs and is able to detect all the potential faults (*Mathur, 2013*; *König, 2012*). Therefore, we use W to demonstrate the effectiveness of the proposed mutant selection method. Table 1 compares the fault detection capabilities of the common test generation methods. Figure 2 is the comparison of the same methods in terms of the fault detection and test suite lengths.

W method consists of two parts. These are the W-set section for testing the next state and the transition coverage set (TCS) for testing the transition (output) between the two states. The transition coverage set is created to visit all transitions. To do this, the tree is obtained from the graph. paths leading to all ends of this tree are found. The set of input values for these paths creates the transition coverage set. Refer to Chows's study (*Chow, 1978*) for detailed information.

The W method consists of two parts, the W-set section for testing the next state and the transition coverage set (TCS) for testing the transition between the two states. TCS is created to visit all transitions. To do this, the tree is obtained from the graph. Paths

**Table 1  Fault detection capability of the presented methods (*König, 2012*).**

| – | Output faults | Transition faults | Extra state faults | Missing state faults |
|---|---|---|---|---|
| Transition Tour | always | – | – | – |
| DS-Method | always | always | – | always |
| W-Method | always | always | always | always |
| UIO-Method | always | almost | almost | almost |

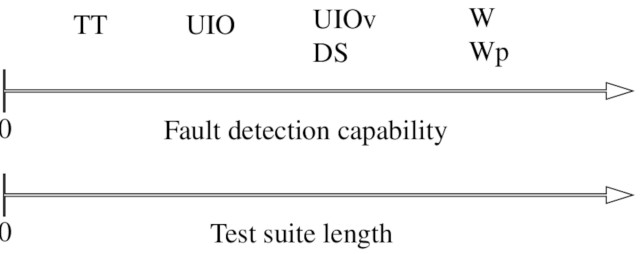

**Figure 2  Fault detection capability versus test suite length of the presented methods.**

leading to all ends of this tree are found. The set of input values for these paths creates the transition coverage set. Refer to Chows's study (*Chow, 1978*) for detailed information.

In this study, the TCS created from the sample FSM shown in Fig. 1 is as follows:

$\{1, 0, 10, 11, 00, 01, 011, 010\}$

The W-set created from this FSM is

$\{11, 1\}$.

By the W method, the test suite is the product of TCS and W-set:

$$W\ TestSuite = TCS \times W\ set = \{1, 0, 10, 11, 00, 01, 011, 010\} \times \{11, 1\} =$$
$$\{11, 1, 111, 11, 011, 01, 1011, 101, 1111, 111,$$
$$0011, 001, 0111, 011, 01111, 0111, 01011, 0101\}$$

## MUTATION TESTING

Starting point of mutation testing is based on the idea of using artificial errors (*Papadakis & Just, 2017*). Mutation testing assumes that small changes in programs are sufficient to reveal complex errors. In the mutation test, the original model is systematically mutated using certain error assumptions. The purpose here is to represent a potential programmer error. Thus, a small change (mutation) is made on the source code. This type of change is based on well-defined mutation operators which can imitate typical programming errors or which can enable generation of highly successful tests.

The most common usage areas of mutation are related to the evaluation of the adequacy of the test suites which provides guidance for test cases (*Zhang et al., 2019*). Mutation analysis is also used in design models, system properties, and system interfaces.

When its principle is examined, it may be seen that the mutation operator replaces an arithmetical operator in the original program with another operator. If a test set can separate a mutant from the original program, it is stated that the mutant is killed. Otherwise, the mutant is called a live mutant. A mutant can remain alive because it is equivalent to the original program, or the test set is insufficient to kill the mutant. If a test suite is not sufficient, test cases can be added to kill the live mutant. The adequacy of a test suite by mutation testing is determined by the ratio of the number of mutants killed to the number of non-equivalent mutants. This rate is also called the mutation score.

$$if\,(a < b)\{\ldots\} \quad \rightsquigarrow \quad if\,(a > b)\{\ldots\}$$

Each modified version of a program, as in the equation above, is called a mutant. If a test case can detect a certain mutant, it is stated that mutant is killed by this test case. The performance of test suites is determined as per the killing rate of mutants being created by means of mutation operators. Mutation testing generally has many advantages, but completing analysis for huge models require a significant amount of time and space because of the large number of mutants present. To avoid this situation, mutant reduction methods are used which limit the number of mutants and enable the analysis to be completed within a reasonable time and space.

There are various mutant reduction methods. Among these, the most commonly used is the random mutant reduction method, a method based on the random selection of sub-sets related to mutants. It is based on the assumption that mutants are identical with respect to the probability of their occurrence. Within frame of this assumption, it is a very practical mutant reduction method. However, this assumption generally does not reflect the truth because the killing of mutants by a test suite is mostly not identical.

When FSMs (Finite State Machine) are tested, the mutation testing technique is frequently preferred. However, while there are many studies related to mutant reduction in the literature, there are few studies related to FSM-based mutant reduction. Existing studies are generally focused on mutation operators being required for mutant generation. In contrast, in this study, a new FSM-based mutation reduction method was proposed because, in huge FSMs, the number of mutants being produced is too great, and this problem is eliminated by reducing mutants. In this study, it was aimed to analyze the regions having the highest probability of fault. Social network analysis centrality criteria were used to identify regions with the highest probability of fault. Hence, with reduced numbers of mutants, the application of mutation testing will be made practical.

As stated by *Fabbri et al. (1994)*, potential mutation operators for FSMs are as follows: Arc-missing, Wrong-starting-state (default state), Event-missing, Event-exchanged, Event-extra, State-extra, Output-exchanged, Output-missing, and Output-extra. According to another article in 2009 (*Li, Dai & Li, 2009*), mutation operators are: Reverse of Transition (ROT), Missing of Transition (MOT), Redundant of Transition (DOT), Change of Input (COI), Missing of Input (MOI), Change of Output (COO), Missing of Output (MOO), End State Changed (ESC), End State Redundant (ESR), Start State Changed (SSC), and Start State Redundant (SSR). To the best of our knowledge, there is no study on prioritizing mutants for reduction.

# CREATION OF MUTANTS BY USING CENTRALITY CRITERIA IN SOCIAL NETWORK ANALYSIS

In the study, a method that can increase efficiency by reducing time and space complexity was proposed. To make the mutation analysis more efficient, instead of creating mutants from the whole model, mutants were created from the major assumed regions; thus, mutation analysis became more effective with a relatively small set of mutants.

To achieve this, the centrality criteria in social network analysis were used. Using centrality criteria, firstly, the central states and transitions in the model were determined. Mutants were created from the determined states and transitions. The centrality criteria used were degree centrality, eigenvector centrality, closeness centrality, and betweenness centrality. In addition, a clustering coefficient was used.

In the mutation analysis, six mutant clusters were created. In the first set, random mutants were generated from the entire model. In other clusters, mutants were created using degree centrality, eigenvector centrality, closeness centrality, betweenness centrality, and clustering coefficient. Finally, these six methods were analyzed with test sets created with the W method.

As a result of the evaluations, it was determined that the proposed mutant reduction method had a higher performance in terms of error capture. It was been observed that the proposed method reduced the cost of mutation analysis.

## Centrality criteria in social network analysis

Social network analysis can be considered as the measurement, display, and analysis of relationships and currents between individuals, organizations, and groups. In such networks, nodes are individuals and groups, and edges refer to the relationship or current between nodes. Social network analysis–which includes such topics as sociology, mathematics, computer science, and statistics–emerges as an interdisciplinary field. Social network analysis, which is based on sociology and mathematics, is a method used by many disciplines today.

Social Network Analysis has become an interesting subject in mathematics and engineering in recent years. As far as we know, there no study in the literature on FSM-based mutant reduction using social network analysis.

Social network analysis can easily identify the most important actors and groups of a structure. In social network analysis, centrality criteria are used to identify the most important actors or groups in a structure. In this context, there are various centrality criteria, and the most often used of these are degree, closeness, betweenness, and eigenvector centrality.

**Degree Centrality:** With the simplest measure, if one node has many connections with other nodes, this node is centrally located in the network. To calculate the degree centering of the x node in a network, the following formula is used:

$$C_d(x) = \frac{c_d(x)}{n-1}. \tag{1}$$

In the above formula, the $c_d(x)$ as node degree and n as the number of nodes is used.

**Closeness Centrality** (*Freeman, 1978*; *Wasserman & Faust, 1994*): This refers to the closeness degree of a node, directly or indirectly, to the other nodes in the network. Closeness reflects the node's ability to access information and how fast a node can connect to other nodes in the network.

$$C(u) = \frac{n-1}{\sum_{v=1}^{n-1} d(v, u)}. \tag{2}$$

In the equation, $d(v, u)$ represents the shortest path between $v$ and $u$ and $n$ is the number of nodes that can reach $u$.

**Betweenness Centrality** (*Brandes, 2001*): This is the degree of a node being among other nodes in the network and indicates to what extent a node is in direct connection with nodes that are not directly linked to each other. In other words, it refers to the extent to which a node takes on the role of a bridge.

$$c_B(v) = \sum_{s,t \in V} \frac{\sigma(s,t|v)}{\sigma(s,t)}. \tag{3}$$

Here, $V$ represents the set of nodes, $\sigma(s,t)$ is the sum of the shortest paths between $s, t$, and $\sigma(s,t|v)$ is the sum of the shortest paths between $s, t$, which inherit the $v$ node.

**Eigenvector Centrality** (*Bonacich, 1987*): This assumes that all connections are not equal and that effective nodes are also transmitting effects to less effective nodes with whom they are linked. If a node has few high-quality connections, this node is more effective in terms of eigenvector centrality than a node with a large number of average connections.

$$Ax = \lambda x. \tag{4}$$

If a scalar $\lambda$ has a nonzero vector $x$ such as $Ax = \lambda x$, it is called the eigenvalue of A. Such a vector $X$ is called an eigenvector corresponding to $\lambda$.

Apart from these, it is also possible to obtain information about centrality by using the cluster coefficient.

**Clustering Coefficient** (*Saramäki et al., 2007*; *Onnela et al., 2005*; *Fagiolo, 2007*): The clustering coefficient is a measure of the probability of two different nodes having a bond with a common node.

$$c_u = \frac{1}{deg^{tot}(u)(deg^{tot}(u) - 1) - 2deg^{\leftrightarrow}(u)} T(u). \tag{5}$$

In the formula above, $T(u)$ represents the number of actual connections between the neighbors of $u$ and $deg^{tot}(u)$ is the sum of the internal and external rating of the node $u$, and $deg^{\leftrightarrow}(u)$ expresses the number of those which have mutual degree.

In this study, mutants created using the above-mentioned centrality criteria were compared with randomly generated mutants. First, the regions where the test sets pass the most were determined by using the centrality criteria of the social network analysis method. The mutants were then generated using mutation operators such as missing of transition (MOT), change of input (COI), change of output (COO), and change of next state (CONS). Then, mutants created according to the centrality criteria and the

ones randomly generated were evaluated with test sets created with the W Method. The results, presented with graphical comparisons as well as the ANOVA Test, showed that the proposed mutant reduction method could provide a significant increase in performance.

## Proposal algorithm

First, the FSM was converted to a directional graph. Since the normal graph structure cannot provide transition between states relative to the input, a data structure expressing the graph was also used.

In the algorithm, the centralities were calculated using social network analysis. Centralities were applied both on state and transition. For transitions, betweenness centrality was applied, but, for states, betweenness, degree, eigenvector and closeness centralities were applied as well as the clustering coefficient.

Each centrality used in the study was calculated separately. The values obtained as a result of the calculation are listed for each centrality specifically. A coefficient was given to the program to determine which conditions would be selected among the values obtained. Based on the given coefficient, the highest value cases were selected.

Example how to calculate degree centrality with FSM is shown in Table 2.

The shortest paths between the states and transitions were first determined to calculate the betweenness and closeness centrality. In Table 3, the shortest paths that exist in the FSM are given.

Using the shortest paths given in Table 3, the calculation of edge betweenness centrality to reduce the transitions is shown in Table 4. Here, the number of shortest paths passing through the selected transitions are formed by dividing the total number of shortest paths available in Table 3.

Using the shortest paths given in Table 3, the calculation of betweenness centrality to reduce the states is shown in Table 5. Here, the number of shortest paths passing through the selected states are formed by dividing the total number of shortest paths available in Table 3.

The calculation of the closeness centrality is shown in Table 6. Here, again using Table 3, the shortest paths from one node to other nodes are found, and the found shortest paths are summed. Then the number of edges entering that node is divided by the totals.

The calculation of eigenvector centrality is explained in Eq. (6). For detailed information, see (*Newman, 2018*).

$$\begin{bmatrix} 0 & 1 & 1 & 0 \\ 1 & 0 & 1 & 0 \\ 0 & 1 & 0 & 1 \\ 0 & 1 & 1 & 0 \end{bmatrix} \cdot \begin{bmatrix} START \\ state1 \\ state2 \\ state3 \end{bmatrix} = \lambda \cdot \begin{bmatrix} START \\ state1 \\ state2 \\ state3 \end{bmatrix} \Longrightarrow \begin{bmatrix} START \\ state1 \\ state2 \\ state3 \end{bmatrix} = \begin{bmatrix} 0.31622776601683794 \\ 0.6324555320336759 \\ 0.6324555320336759 \\ 0.31622776601683794 \end{bmatrix}$$

(6)

Clustering coefficient was calculated as in Formula (5). These operations are shown in Table 7.

Within the scope of social network analysis, fewer mutants are created with the states and transitions selected by calculating the centrality values.

The complexity analysis values of the proposed algorithms are given in Table 8.

**Table 2  Degree centrality calculation.**

| States | Degree | Degree centrality |
| --- | --- | --- |
| START | 3 | 3 / 3 = 1 |
| state1 | 5 | 5 / 3 = 1.666 |
| state2 | 3 | 5 / 3 = 1.666 |
| state3 | 5 | 3 / 3 = 1 |

**Table 3  Shortest path in FSM.**

| The shortest paths |
| --- |
| START - state1 |
| START - state2 |
| START - state2 - state3 |
| state1 - START |
| state1 - state2 |
| state1 - state2 - state3 |
| state2 - state1 |
| state2 - state1 - START |
| state2 - state3 |
| state3 - state1 |
| state3 - state2 |
| state3 - state1 - START |

**Table 4  Edge betweenness centrality calculation.**

| Transitions | Edge betweenness | Edge betweenness centrality |
| --- | --- | --- |
| START - state1 | 1 | 1/12 = 0.083 |
| START - state2 | 2 | 2/12 = 0.166 |
| state1 - START | 3 | 3/12 = 0.25 |
| state1 - state2 | 2 | 2/12 = 0.166 |
| state2 - state1 | 2 | 2/12 = 0.166 |
| state2 - state3 | 3 | 3/12 = 0.25 |
| state3 - state1 | 2 | 2/12 = 0.166 |
| state3 - state2 | 1 | 1/12 = 0.083 |

## EVALUATIONS

In this section, the outputs obtained as a result of comparing the mutant reduction technique proposed in the study and the mutant reduction technique in the literature with the W method are presented. The reason for using the W method here was because of the ability of the W method to catch all potential errors, as mentioned earlier. According to the data obtained as a result of the comparison, the success of the proposed mutant reduction methods was observed.

The transition and states in FSM, which are determined by using social network analysis centrality criteria, were mutated. Only the betweenness centrality was used for the

**Table 5  Betweenness centrality calculation.**

| States | Betweenness | Betweenness centrality |
|---|---|---|
| START | 0 | 0 |
| state1 | 2 | 2 / 12 = 0.16 |
| state2 | 2 | 2 / 12 = 0.16 |
| state3 | 0 | 0 |

**Table 6  Closeness centrality calculation.**

| | START | state1 | state2 | state3 | Total | Closeness centrality |
|---|---|---|---|---|---|---|
| **START** | 0 | 1 | 1 | 2 | 4 | (1 - 1) / 4 = 0 |
| **state1** | 1 | 0 | 1 | 2 | 4 | (3 - 1) / 4 = 0.5 |
| **state2** | 2 | 1 | 0 | 1 | 4 | (3 - 1) / 4 = 0.5 |
| **state3** | 2 | 1 | 1 | 0 | 4 | (1 - 1) / 4 = 0 |

**Table 7  Calculation of clustering coefficient.**

| States | Clustering coefficient |
|---|---|
| START | $\frac{8}{(3\times(3-1)-2\times1)\times2} = 1$ |
| state1 | $\frac{16}{(5\times(5-1)-2\times2)\times2} = 0.5$ |
| state2 | $\frac{16}{(5\times(5-1)-2\times2)\times2} = 0.5$ |
| state3 | $\frac{8}{(3\times(3-1)-2\times1)\times2} = 1$ |

transitions. As for the states, since there are many centrality criteria in the literature, each one was tried separately. The centrality criteria we used in the proposed reduction methods are shown in Table 9.

In comparing the mutant reduced with the method available in the literature and the mutant reduced with the method proposed in the study, the following mutation operators were used: Missing of transition (MOT), change of input (COI), change of output (COO), and change of next state (CONS).

Test cases are checked against the mutants whether they are killed or not. Mutation analysis is generally beneficial in assessing the effectiveness of test cases. The mutant kill ratio, that is, the number of killed mutants over the entire mutant set, is a common performance metric. Since the W method provides the most comprehensive test suite for a given FSM, the expected mutant kill ratio is 100%. In fact, our mutant reduction method aimed to select the most influential mutants. Those mutants are more likely to be killed by a test suite, and they demonstrate a better performance in revealing the faults. Therefore, we define a mutant importance metric as given in Definition 7.1.

**Definition 7.1  (Mutant Importance Metric):**   *Let $M = \{m_1, m_2, m_3...m_n\}$ be a set of mutants. The performance metric of mutant  iis,*

$$\mathcal{I}(m_i) = Number\ of\ test\ cases\ killing\ m_i$$

**Table 8   Complexity analysis of centrality algorithms.**

| Algorithm | Complexity | Reference |
|---|---|---|
| Degree centrality | O(V) | – |
| Eigenvector centrality | O(V + E) | (*Bonacich, 1987*; *Newman, 2018*) |
| Closeness centrality | O(V*(V + E)) | (*Freeman, 1978*; *Wasserman & Faust, 1994*) |
| Betweenness centrality | $O(V^3)$ | (*Brandes, 2001*; *Brandes, 2008*) |
| Edge betweenness centrality | $O(V^3)$ | (*Brandes, 2001*; *Brandes, 2008*) |
| Clustering coefficient | $O(V^3)$ | (*Saramäki et al., 2007*; *Onnela et al., 2005*; *Fagiolo, 2007*) |

**Table 9   Centrality criteria used in proposed reduction methods.**

| Proposed algorithm | Transitions | States |
|---|---|---|
| Closeness Reduction | Edge Betweenness Centrality | Closeness Centrality |
| Betweenness Reduction | Edge Betweenness Centrality | Betweenness Centrality |
| Eigenvector Reduction | Edge Betweenness Centrality | Eigenvector Centrality |
| Degree Reduction | Edge Betweenness Centrality | Degree Centrality |
| Clustering Reduction | Edge Betweenness Centrality | Clustering Coefficient |

The performance of the entire mutant set can be expressed as the average importance as defined in Definition 7.2.

**Definition 7.2 (Total Importance Metric):**   *Let $M = \{m_1, m_2, m_3...m_n\}$ be a set of mutants. The performance of the mutant set  M is,*

$$\mathcal{I}(M) = \mathbf{E}_i[\mathcal{I}(m_i)] = \frac{1}{n}\sum_{i=1}^{n}\mathcal{I}(m_i)$$

FSMs are randomly generated by the KISS Generator v0.8 software tool, changing the state, output, and input values separately. Each FSM is produced five times, and the mean values of outcomes are calculated. This procedure is repeated 300 times, which results in 1,500 FSMs in total. According to the existing similar studies, this amount seems sufficient and cannot be considered a serious threat to validity. Similarly, random generation of values eliminates another validation threat. FSMs are *reduced*, *deterministic*, *fully correlated*, and *fully specified*, as required by the W method.

In the evaluations, $\mathcal{I}(M)$ is measured against three different FSM parameters: (i) Number of states, (ii) Number of inputs and (iii) Number of outputs. We compare the original set of mutants and its subset selected by the proposed methods.

In Fig. 3, the mutants created according to the social network analysis centrality criteria were compared with randomly generated mutants. While the values increased linearly in the first graph, the values remained flat in the second and third graphs. On the other hand, there was a significant difference between mutants created according to the social network analysis centrality criteria and the randomly generated mutants. The ANOVA test also confirmed this difference. Based on this difference, it is possible to say that the proposed method had a higher performance in the MOT operator. Furthermore, according to the ANOVA test, there was no significant difference between the methods created with the social network analysis centrality criteria.

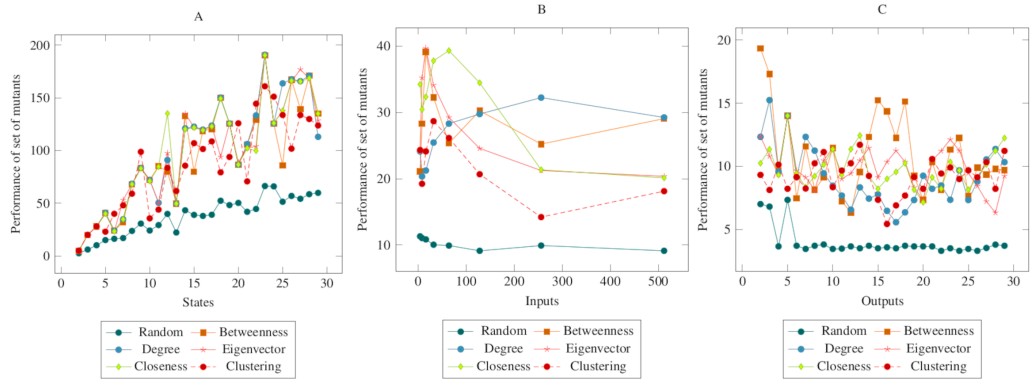

**Figure 3   Comparisons for missing of transition (MOT) operator.** (A) State, (B) Input, (C) Output.

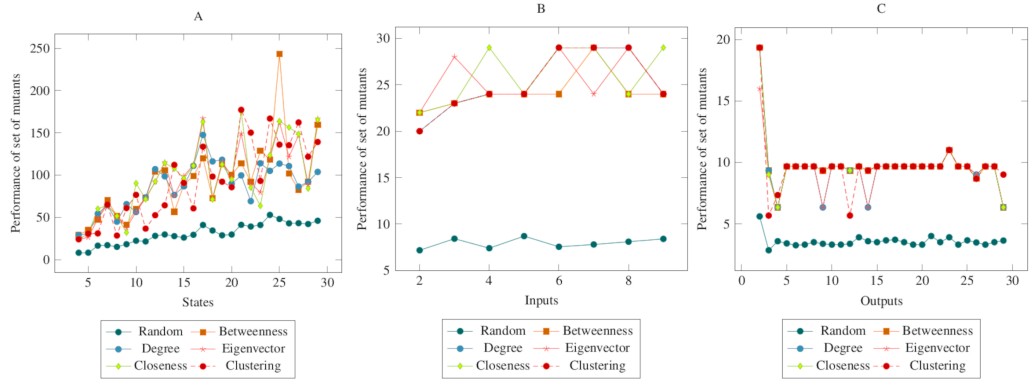

**Figure 4   Comparisons for change of output (COO) operator.** (A) State, (B) Input, (C) Output.

Figure 4 showed similarities to Fig. 3. However, a more distinct difference was observed in the inputs. At the same time, it can be said that the centrality-based reductions showed more similar results in the graph created according to the output. Also, according to the ANOVA test, there was a significant difference between the centrality-based mutant reduction methods and the randomly selected mutant reduction method. Based on this difference, it is possible to say that the proposed method showed a higher performance in the COO operator. Furthermore, according to the ANOVA test, there was no significant difference between the methods created with the social network analysis centrality criteria.

Figure 5 showed similarities with Figs. 3 and 4. However, in comparison with the output graph, the difference between the random mutant reduction method and the proposed method in the state and the input graph had relatively decreased. However, according to the ANOVA test, it was possible to say that the mutant reduction methods created with centrality criteria differed significantly from the randomly generated mutant reduction method. Based on this difference, it is possible to say that the proposed method shows a higher performance in the COI operator. On the other hand, according to the ANOVA
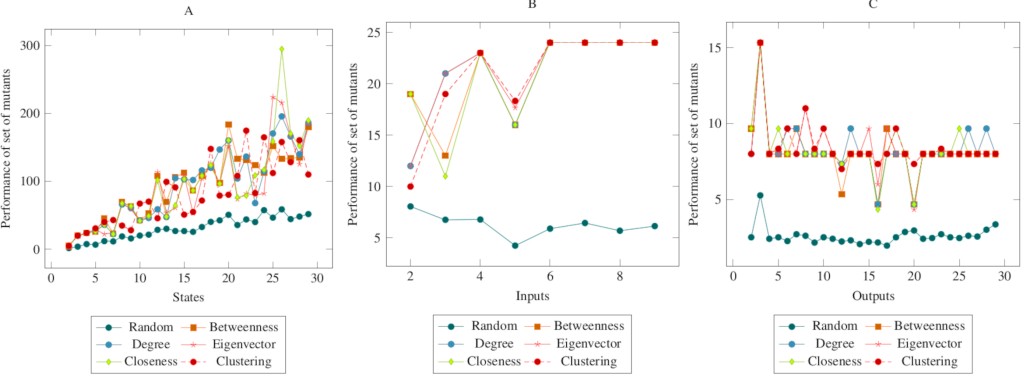

**Figure 5** **Comparisons for change of input (COI) operator.** (A) State, (B) Input, (C) Output.

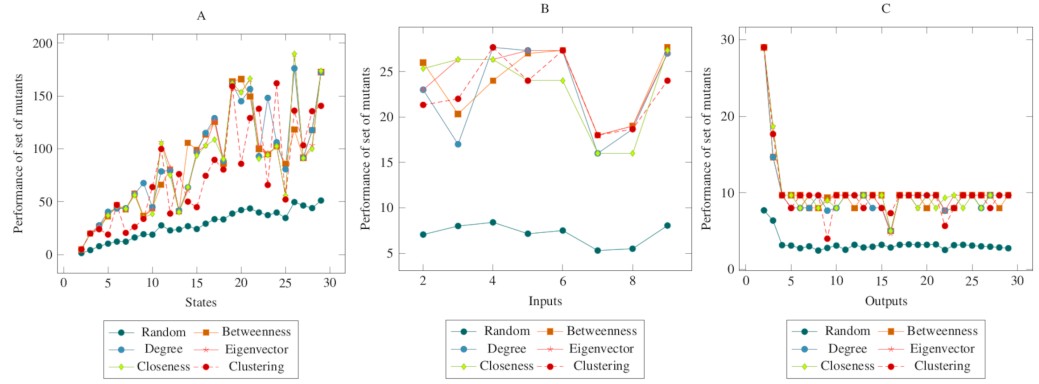

**Figure 6** **Comparisons for change of next state (CONS) operator.** (A) State, (B) Input, (C) Output.

test, there was no significant difference between the methods created with social network analysis centrality criteria.

The values in Fig. 6 showed similarities with the values in the previous figures. In other words, there was a difference between the centrality-based mutant reduction methods and the random mutant reduction methods. The ANOVA values also supported this situation. Based on this difference, it is possible to say that the proposed method showed a higher performance in the CONS operator. On the other hand, according to the ANOVA test, there was no significant difference between the methods created with social network analysis centrality criteria.

In Fig. 7, the performance of mutants is shown collectively according to the change of input, output, and subsequent state with a lack of transition. Accordingly, the total values showed similarities with the values in the figures given separately above. The ANOVA values also supported these indicators. Based on this difference, it is possible to say that the proposed method showed a higher performance in all of the MOT, COI, COO and CONS operators. On the other hand, according to the ANOVA test, there was no significant difference between the methods created with social network analysis centrality criteria.

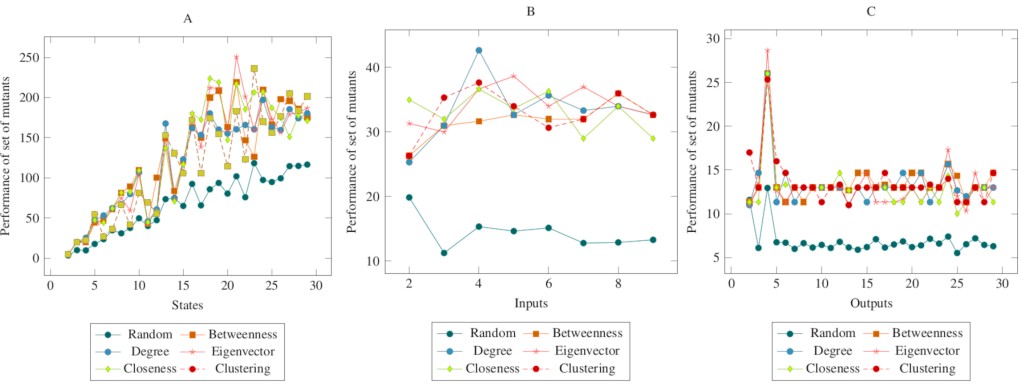

**Figure 7** **Comparisons for all mutation operators, CONS, MOT, COI and COO.** (A) State, (B) Input, (C) Output.

When looking at the graphs in general, it can be said that the main reason for the difference between the random mutant reduction method and the proposed mutant reduction methods was "betweenness centrality applied for transitions."

## DISCUSSION

For evaluation purpose, large amounts of FSMs have been produced in order to make the results become more consistent. According to the literature, it is possible to state that number of mutants created from FSM is sufficient for such a study, and that it is possible to generalize the study. The success of proposed mutant reduction methods have been verified by being compared with the random mutant reduction method. To measure meaningful success performance in the study, ANOVA and a graphical method were chosen. According to the evaluation results, it is possible to state that the proposed mutant reduction method plays an active role in reducing number of mutants that are produced from FSM without having an important reduction in the capacity to catch faults in FSM.

In this study, two major parameters, transition and state reduction ratios, were used for mutant reduction by using social network analysis centrality criteria. These parameters were selected intuitively. By investigating these parameters further in following studies, it would also possible to find more appropriate values.

The scarcity of research in the literature on mutant reduction in FSMs allows this matter to be developed with different approaches. In this study, social network analysis was used in the mutant reduction method, the first such experiment, as far we know. From this point of view, social network analysis could also be used in different types of research on software.

## CONCLUSION

In our study, a method was developed for the reduction of the mutant set for mutation analysis, which is an analysis technique widely used in circuit and software tests.

In this context, first, important states and transitions in FSM were determined by using the social network analysis centrality criteria. Thus, instead of an entire model, mutants were created based on the detected regions, and the number of mutants was reduced. In the study, clustering coefficient criteria along with degree, eigenvector, closeness, and betweenness centrality were used to determine the states and transitions.

Mutation operators such as missing of transition (MOT), change of input (COI), change of output (COO), and change of next state (CONS) were used to produce the mutants. Then, mutants randomly reduced and mutants reduced according to the centrality criteria were evaluated with test sets created with the W method. The results were presented with graphical comparisons and the ANOVA Test.

Looking at the averages of the study data, it can be seen that the closeness reduction yielded slightly better results than other proposed mutant reduction methods. When looking at the ANOVA results and graphics in general, it can be said that the main reason for the difference between the random mutant reduction method and the proposed mutant reduction methods is betweenness centrality applied for transitions. As a result, it was observed that the mutants selected with the proposed reduction technique performed better, and the proposed method reduced the cost of the mutation analysis.

In this article, the method developed provided alternative suggestions on how social network analysis could be applied to software testing and related issues. In addition, it was thought that social network analysis could provide many advantages in software tests since its implementation is practical.

### Funding
The author received no funding for this work.

### Competing Interests
The authors declare there are no competing interests.

### Author Contributions
- Savaş Takan conceived and designed the experiments, performed the experiments, analyzed the data, performed the computation work, prepared figures and/or tables, authored or reviewed drafts of the paper, and approved the final draft.

### Data Availability
Code is available at GitHub: https://github.com/savastakan/sna_mut/.

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
