# Peer review of "Creation of mutants by using centrality criteria in social network analysis"

_PeerJ Computer Science, doi:10.7717/peerj-cs.293_

## Round 0.1 · original submission · Major Revisions

From the comments of the reviewers, the paper has some importance and contribution. However, there are some disadvantages, especially for presentions. The authors should improve the paper according to the comments of the reviewers.

Reviewer 1 ·

Basic reporting

The English in the paper has to be improved substantially, as it is very difficult to understand and follow the arguments included in the paper. Table 7, for example, is not in English but Turkish.

Sufficient background information and references have been provided but again due to the bad use of the language it is not very easy to follow them.

The structure needs to be revised as well, just Table 7 shows the deficiency in that respect.

The paper is self contained and relevant results to the hypothesis have been obtained. But the presentation of the results (e.g. Figure 3) is not very clear.

Otherwise the paper is written in accordance with formal rules.

Experimental design

The paper proposes a mutant selection method using different criteria derived from social network analysis. Mutants are artificially introduced faults used to test the efficiency of tests developed for Finite state machines. It is important to generate include these into systems efficiently in order to measure the performance of the test sets.

The question adequately designed and and has some importance and contribution, but in my opinion it is not a very significant one.

The author has performed rigorous analysis, and results reflect these although not very well presented.

These results would be useful for future researchers who may want to compare their results with others.

Validity of the findings

As stated before there is some novelty in the findings but the impact may be questioned.

The author presented the study in detail but again these are veiled due to language usageç

The conclusion may be improved, but contents are appropriate for the findings.

Additional comments

None

·

Basic reporting

The work presented in this study is qualified for publication, however, there are some errors related to the spelling and structure of the work, please modify them.

Experimental design

The experiments are designed properly.

Validity of the findings

The data provided by the user is appropiately stated.

Additional comments

1. Page 8, please have a check at Table 4.
2. Could you please check the names of Table 7?
3. Please check Table 9, why the name of the table is below the table while others are at the above?

---

## Round 0.2 · accepted · Accept

From the response of the reviewer, I believe the paper has been revised well.

Reviewer 1 ·

Basic reporting

The paper has been significantly improved and became more readable hence understandable. The tables and figures were also corrected. There are still some minor English mistakes which may be useful to revise.

Experimental design

The aims and scope has been more clearly defined.

Validity of the findings

The findings are clearer after the revision.

Additional comments

None